# Large-Scale Profiling of Extracellular Vesicles Identified miR-625-5p as a Novel Biomarker of Immunotherapy Response in Advanced Non-Small-Cell Lung Cancer Patients

**DOI:** 10.3390/cancers14102435

**Published:** 2022-05-14

**Authors:** Francesco Pantano, Francesca Zalfa, Michele Iuliani, Sonia Simonetti, Paolo Manca, Andrea Napolitano, Simone Tiberi, Marco Russano, Fabrizio Citarella, Simone Foderaro, Elisabetta Vulpis, Alessandra Zingoni, Laura Masuelli, Roberto Bei, Giulia Ribelli, Marzia Del Re, Romano Danesi, Bruno Vincenzi, Giuseppe Perrone, Giuseppe Tonini, Daniele Santini

**Affiliations:** 1Medical Oncology Department, Campus Bio-Medico University of Rome, 00128 Rome, Italy; f.pantano@policlinicocampus.it (F.P.); s.simonetti@unicampus.it (S.S.); a.napolitano@unicampus.it (A.N.); m.russano@policlinicocampus.it (M.R.); f.citarella@unicampus.it (F.C.); s.foderaro@unicampus.it (S.F.); ribelli.giuly@live.it (G.R.); b.vincenzi@policlinicocampus.it (B.V.); g.tonini@policlinicocampus.it (G.T.); d.santini@policlinicocampus.it (D.S.); 2Pathology Department, Campus Bio-Medico University, 00128 Rome, Italy; f.zalfa@unicampus.it (F.Z.); g.perrone@policlinicocampus.it (G.P.); 3Department of Medical Oncology, Fondazione IRCCS Istituto Nazionale dei Tumori, 20133 Milan, Italy; paolo.manca@istitutotumori.mi.it; 4Department of Molecular Life Sciences and Swiss Institute of Bioinformatics, University of Zurich, Winterthurerstrasse 190, 8057 Zurich, Switzerland; simone.tiberi@uzh.ch; 5Laboratory Affiliated to Istituto Pasteur Italia-Fondazione Cenci Bolognetti, Department of Molecular Medicine, Sapienza University of Rome, 00161 Rome, Italy; elisabetta.vulpis@uniroma1.it (E.V.); alessandra.zingoni@uniroma1.it (A.Z.); 6Department of Experimental Medicine, Sapienza University of Rome, 00161 Rome, Italy; laura.masuelli@uniroma1.it; 7Department of Clinical Sciences and Translational Medicine, University of Rome “Tor Vergata”, 00133 Rome, Italy; bei@med.uniroma2.it; 8Unit of Clinical Pharmacology and Pharmacogenetics, Department of Clinical and Experimental Medicine, University of Pisa, 56126 Pisa, Italy; marzia.delre@unipi.it (M.D.R.); romano.danesi@unipi.it (R.D.); 9UOC of Oncology-ASL Latina-Distretto 1, University of Rome “Sapienza”, 04011 Aprilia, Italy

**Keywords:** immune checkpoint inhibitors, non-small-cell lung cancer, extracellular vesicles, microRNAs

## Abstract

**Simple Summary:**

Immune checkpoint inhibitors (ICIs) have revolutionized the treatment of advanced non-small-cell lung cancer (NSCLC) leading to substantial improvement in survival time and quality of life. Nevertheless, the clinical benefit of treatment is still limited to a minority of patients, reflecting the need to identify novel noninvasive biomarkers to improve patient selection. Currently available markers such as PD-L1 expression have important limitations. In this study, we focused on extracellular vesicles (EV)-associated miRNAs produced by cancer cells and their microenvironment that can be easily detected in blood. In particular, after a large-scale screening of 799 EV-miRNAs, we identified EV-miR-625-5p as a novel independent biomarker of response and survival in ICI-treated NSCLC patients, in particular in patients with PD-L1 expression ≥ 50%. EV-miR-625-5p integrated with PDL-1 test could allow the clinician to identify in advance patients that would benefit from ICIs.

**Abstract:**

Immune checkpoint inhibitors (ICIs) are largely used in the treatment of patients with advanced non-small-cell lung cancer (NSCLC). Novel biomarkers that provide biological information that could be useful for clinical management are needed. In this respect, extracellular vesicles (EV)-associated microRNAs (miRNAs) that are the principal vehicle of intercellular communication may be important sources of biomarkers. We analyzed the levels of 799 EV-miRNAs in the pretreatment plasma of 88 advanced NSCLC patients who received anti-PD-1 therapy as single agent. After data normalization, we used a two-step approach to identify candidate biomarkers associated to both objective response (OR) by RECIST and longer overall survival (OS). Univariate and multivariate analyses including known clinicopathologic variables and new findings were performed. In our cohort, 24/88 (27.3%) patients showed OR by RECIST. Median OS in the whole cohort was 11.5 months. In total, 196 EV-miRNAs out 799 were selected as expressed above background. After multiplicity adjustment, abundance of EV-miR-625-5p was found to be correlated with PD-L1 expression and significantly associated to OR by RECIST (*p* = 0.0366) and OS (*p* = 0.0031). In multivariate analysis, PD-L1 staining and EV-miR-625-5p levels were constantly associated to OR and OS. Finally, we showed that EV-miR-625-5p levels could discriminate patients with longer survival, in particular in the class expressing PD-L1 ≥50%. EV-miRNAs represent a source of relevant biomarkers. EV-miR-625-5p is an independent biomarker of response and survival in ICI-treated NSCLC patients, in particular in patients with PD-L1 expression ≥50%.

## 1. Introduction

Immune checkpoint inhibitors (ICIs) have dramatically changed the therapeutic landscape for patients with non-small-cell lung cancer (NSCLC) [1]. However, only a subset of patients responds to ICIs, and biomarkers currently used, such as programmed cell death ligand 1 (PD-L1) and tumor mutation burden (TMB), exhibit limited predictive capabilities [2]. Cancer cells and their microenvironment produce heterogeneous mixtures of extracellular vesicles (EVs), namely exosomes and microvesicles (MVs), that can be detected in body fluids, including blood [3]. EVs released by cancer cells can suppress the immune-system response, inactivating T lymphocytes or natural killer cells, as well as promoting differentiation of regulatory T lymphocytes and tumor growth [4]. An important breakthrough was the discovery of the presence of nucleic acids such as mRNA and miRNA in EVs. In particular, miRNAs, a class of small, single-stranded noncoding RNAs, have emerged as key players in modulating cancer cell phenotype, and more recently, as crucial regulators of innate and adaptive immune responses by negatively regulating the expression of key regulators of developmental checkpoints [5,6].

From this perspective, circulating EV-associated miRNAs (EV-miRNAs) could provide relevant information regarding not only cancer cell biology but also tumor microenvironment, including tumor–immune system interactions. In particular, emerging data suggest that tumor-derived EVs are enriched in immunosuppressive proteins or in microRNAs targeting suppressive pathways in recipient cells contributing to reprogramming the tumor microenvironment into a cancer-promoting milieu [7,8].

Several groups investigated the role of circulating miRNAs as predictors of clinical outcome in NSCLC patients undergone to ICI treatment [7,8,9,10]. Halvorsen et al. identified a serum miRNA signature associated with survival in nivolumab-treated NSCLC patients [9]; Boeri et al. applied a prognostic plasma immune-related miRNA-signature classifier, previously identified, in NSCLC patients prior to the initiation of ICI therapy [10] finding a correlation with clinical outcome. Moreover, two small-sized studies showed a significant association between baseline levels of some plasma EV-miRNAs and response to ICI therapy [11,12].

Here, we screened EV-miRNAs’ profile in plasma of 88 advanced NSCLC patients who underwent anti-PD-1/PD-L1 therapy as single agent to identify potential novel biomarkers of response to ICIs.

## 2. Patients and Methods

### 2.1. Patients

#### 2.1.1. Sample Size

Objective response (OR) is a direct measure of a drug antitumor activity that can be evaluated in single-arm studies. It represents a reliable clinical endpoint to identify candidate biomarkers of biological significance, as it is less influenced than time-dependent outcomes from other clinical variables and can hence be attributable directly to the drug, not the natural history of the disease. However, OR is considered a relatively poor surrogate of overall survival (OS) in patients treated with ICIs [13].

To identify EV-miRNAs of both biological and clinical interest, we designed our study in two steps. In the first step, we considered two groups (responder; nonresponder according RECIST 1.1 criteria) designed to identify EV-miRNAs significantly associated after false discovery rate (FDR) correction to OR during treatment with ICIs. The following parameters were used: G0 (estimating a number of undifferentially detectable EV-miRNAs) = 250; E(R0) (mean number of false positives) = 2; expected differential expression between case and control conditions of |μ1| = 0.5 on a log-2 scale; anticipated experimental error standard deviation (σ) = 0.70 on a log-2 scale; standard deviation of the difference in log-expression between treatment and control conditions (σd) = √2σ = √2(0.70)= 0.9899; ratio |µ1|/σd = 1.000/0.9899 = 1.010. For these specifications, 24 samples for each group were needed, with a noncentrality parameter ψ1 = 48(1.010)2 = 48.9648 [14].

Secondly, expression levels of miRNA(s) successfully identified in the first step were dichotomized according to an optimized cutoff value (based on distribution and sensitivity/specificity method). With a hazard ratio (HR) threshold of 0.50/2.00, a value for alpha of 0.05 (one-sided), a desired power of 80% and estimating an allocation ratio from the optimal dichotomization process from 1:1 to 1:3 (i.e., with the group with the lower number of patients between 50% and 25% of the total sample size), the total number of required events (deaths) ranged from 51 to 69 [15].

Following the most conservative estimates, we therefore planned to continue enrollment and follow-up until obtaining a cohort of informative patients with at least 24 OR and 69 OS events.

#### 2.1.2. Study Design

A consecutive series of 218 NSCLC patients was administered with anti-PD1 from 01-2018 to 02-2020 and followed up until 05-2021 at Campus Bio Medico of Rome University Hospital (Rome, Italy). A total of 48 patients were excluded from the study because they did not meet inclusion criteria, and 6 because they declined to participate.

The study was conducted in accordance with the principles of the Helsinki Declaration. All experimental protocols were approved by the Internal Review and Ethics Boards of the Campus Bio Medico University Hospital of Rome (Prot. N. 48.17OSS) and all patients provided informed consent.

Plasma samples were all collected prior the first cycle of ICI. A total of 35 out 164 patients’ plasma were excluded for low yield or presence of hemolysis. The patients’ disease had to be measurable per RECIST 1.1 at baseline and had to be periodically evaluated for response to treatment by radiological evaluation (CT scan or PET/CT scan). All patients’ disease progression (PD) had to be demonstrated by radiological evaluation. According to the RECIST 1.1 best response criteria, patients were classified as responders (R), patients with stable disease (SD), and PD. The OR was defined as complete response (CR) or a partial response (PR). Tumor burden calculated as the sum of diameters of all target lesions (unidimensional measurements) for all patients included in the study at every radiological evaluation. Progression-free survival (PFS) was defined as the time from the first infusion of anti-PD1 to the first documented tumor progression. OS was defined as the time from the first infusion of anti-PD1 to death or last news. A total of 21 out of 129 patients with adequate plasma were excluded because they died before first radiological evaluation or were lost to follow up. A total of 20 out of 108 patients included for miRNA expression analysis did not pass Ncounter Quality Check, leading to 88 patients that were included in the final study analysis. Clinical and biological data were collected through a dedicated patient file database. The redaction of the manuscript followed the TRIPOD guidelines for prognostic/predictive studies [16].

##### Inclusion and Exclusion Criteria

The inclusion criteria were:patients who provided written informed consent for the studymale or female subjects 18 years of age, with a performance status of 0–1, with adequate organ function and no signs of active autoimmune diseasepatients treated with anti-PD-1 (pembrolizumab or nivolumab) as monotherapy as first, second or third line of treatment for advanced disease.

The inclusion criteria were: subjects who had an history of invasive malignancy ≤5 years, except for adequately treated basal cell or squamous cell skin cancer or in situ cervical cancer.patients who received prior therapy with an anti-PD-1, anti-PD-L1 or anti-PD-L2 agent or with an agent directed to another coinhibitory T-cell receptordiagnosis of immunodeficiency or received systemic steroid therapyan active infection requiring systemic therapy

### 2.2. Methods

#### 2.2.1. EV miRNA Profiling of Plasma Samples

Whole blood was collected the same day of the first ICI infusion in four 3 mL K2EDTA Vacutainer tubes and the plasma was collected after two centrifugation steps at 1258× *g* for 10 min. EV total RNA isolation was performed using exoRNeasy Serum/Plasma Kit (Qiagen, Hilden, Germany) according to manufacturers’ protocol. miRNA expression analysis was assessed by nCounter Analysis System (NanoString Technologies, Seattle, WA, USA).

#### 2.2.2. Data Normalization

Reporter Code Count (RCC) files generated by nCounter instrument were imported to nSolver 4.0 software (NanoString Technologies Seattle, WA, USA). Quality Check (QC) on Binding Density, Image Quality and Positive Control Linearity as well as Positive Control Limit of Detection and Ligation was performed using the default QC settings. Samples (*n* = 20) that did not pass QC were excluded from the analysis.

In order to identify the pool of miRNAs considered detectable from nCounter platform, raw expression of endogenous miRNAs was compared with that of negative controls. Kruskal–Wallis test followed Dunn’s post test was used to select miRNAs with mean ranks significantly higher compared to the mean ranks of negative controls. 196 miRNAs out 799 were selected as expressed above background (Appendix A).

Several normalization methods were hence explored using NormalyzerDE R package. Normalyzer is implemented in R using Bioconductor packages.

In particular, Global Intensity (GI), Median Intensity (Median), Mean Intensity (Mean), Quantile (preprocessCore package), Variance Stabilizing Normalization (VSN, vsn package), Robust Linear Regression (RLR) and CycLoess (limma package) were explored.

Selection of the optimal normalization method was carried out considering different quantitative and qualitative statistical measures. CycLoess resulted in a normalization method with the lowest Pooled intragroup Coefficient of Variation (PCV), Pooled intragroup Median Absolute Deviation (PMAD) and Pooled intragroup estimate of variance (PEV) (Appendix A). Presence of bias introduced during normalization was hence excluded using qualitative plots checking for skewness (Density Plot), for improvement in whisker alignment and lengths among replicates (RLE Box Plot), checking if variance is independent of mean (MeanSDplots) and checking for replicate clustering and outlier presence (MDS Plots) (Appendix A).

#### 2.2.3. Statistical Analysis

Limma R package was used to identify differentially expressed miRNAs that can discriminate between patients who achieved OR (responders) vs. patients who experienced SD or PD as best response (nonresponders) to ICI treatment. FDR correction for multiple hypothesis testing was applied. OptimalCutpoints R package was used to identify the optimal cutoff to dichotomize patients according hsa-miR-625-5p (miR-625-5p) expression and OR (Appendix A). Optimal cutoff was selected for its ability to best separate density curves of responders and nonresponders. CutoffFinder R package was used to plot, respectively, HRs for OS and PFS and odd ratios for OR of all possible cutoffs) (Appendix A). 

Univariable and multivariable logistic regression models was used to determine Odd Ratios and 95% confidence intervals (CIs) for OR. Survival curves were estimated by the Kaplan-Meier method and compared with the log-rank test (univariate analysis). Univariate HRs were calculated using log-rank method. Multivariable Cox regression model was used to determine HRs and 95% confidence intervals (CIs) for OS and PFS. Variables found to be significantly associated to OR, OS and PFS at the *p* < 0.05 level in the univariate analysis were entered into a multivariate models.

Differences between tumor change at best response and miR-625-5p/PD-L1 status were evaluated using Kruskall–Wallis followed by Dunn test for pairwise comparison with Bonferroni Adjustment. The Spearman rank and Pearson correlation test were employed to examine relationships between tumor change at best response and miR-625-5p expression as continuous variable. 

## 3. Results

### 3.1. Clinicopathological Findings of the Patient Population

From 2018 to 2020, a consecutive series of 218 advanced NSCLC patients treated with anti-PD1 therapy was assessed for eligibility. Of these, 88 were analyzed for pretreatment EV-miRNAs. A flow chart with the reasons for exclusion is provided (Appendix A). In the final cohort of patients, the median follow up was 38 months (95% CI 33—not reached). The median PFS and OS were 6.45 (95% CI 4.70–12.0) and 11.5 months (95% CI 8–16.5), respectively. Among these patients, 35 (39.8%) were treated with nivolumab and 55 (60.2%) with pembrolizumab. Thirty-seven patients (42.0%) received an anti-PD1 therapy as first-line therapy, with 51 (58.0%) in second or third line. Female patients were 27 (30.7%) and 46 patients (52.3%) were less than 70 years old. Sixty-three patients (71.6%) had an adenocarcinoma histology, while the remaining 25 (28.4%) had a squamous cell histology. Forty-four patients (50.0%) had PD-L1 immunohistochemical staining <50%. The complete patient characteristics are summarized in (Table 1).

### 3.2. mir-625-5p Is Associated to OR during Treatment with ICIs

The isolated-EV were characterized through electron microscopy confirming the presence of microvesicles, whose diameter ranges from 130 to 350 nm and exosomes (50–100 nm) (Appendix A) [17]. Differentially expressed miRNAs extracted from EVs were normalized and evaluated based on OR during treatment with ICIs (responder vs nonresponder groups). Ten EV-miRNAs showed a significant difference in expression between groups (*p* < 0.05) before FDR correction. After FDR correction, the only significant differentially expressed miRNA was miR-625-5p (FDR: 0.0366) (Figure 1A). miR-625-5p showed no significant association with all explored clinical variables with the exception of PD-L1 (Appendix A). 

Notably, miR-625-5p levels significantly correlated with tumor size change at best response (Spearman’s ρ = 0.35, *p* = 0.001; Pearson’s r = 0.33, *p* = 0.002) (Figure 1B).

Next, the optimal cutoff value of miR-625-5p to discriminate between groups was evaluated with sensitivity and specificity analyses. A cutoff of 5.47 was selected (Appendix A). Based on this threshold, 55 patients were classified as miR-625-5p High and 33 patients were classified as miR-625-5p Low. The miR-625-5p Low class showed a median tumor size reduction of 30% compared to a median tumor size increase of 18% in the miR-625-5p High class. This difference was highly significant (*p* < 0.001) (Figure 2).

### 3.3. MiR-625-5p Class Is Associated with Survival in NSCLC Patients Treated with ICIs

First, OS and PFS were evaluated in our cohort based on miR-625-5p classes. The median OS in the miR-625-5p Low and High class were respectively 20.0 months (95% CI 13.0–not reached) and 8.0 (95% CI 6.1–11.0) (HR 2.14, 95% CI 1.28–3.58, *p* = 0.0031) (Figure 3A). Similarly, the median PFS in the miR-625-5p Low and High class were, respectively, 13.2 months (95% CI 6.9–27.0) and 4.7 (95% CI 3.1–7.3) (HR 2.04, 95% CI 1.24–3.35, *p* = 0.0046) (Figure 3B).

Next, univariate analyses for OR, PFS and OS with all clinicopathological variables, described in Table 1 and Appendix A, were conducted. Briefly, the variables significantly associated to OR were setting of treatment (*p* < 0.001), type of treatment (*p* = 0.002), PD-L1 staining (*p* < 0.001) and miR-625-5p class (*p* < 0.001) (Appendix A). 

Those significantly associated to OS were setting of treatment (*p* = 0.025), type of treatment (*p* = 0.005), ECOG (*p* = 0.044), tumor burden (*p* = 0.005), presence of liver metastases (*p* < 0.001), PD-L1 staining (*p* < 0.001) and miR-625-5p class (*p* = 0.004) (Appendix A). Lastly, the variables significantly associated to PFS were setting of treatment (*p* = 0.016), type of treatment (*p* = 0.031), tumor burden (*p* = 0.003), presence of liver metastases (*p* = 0.002), PD-L1 staining (*p* < 0.001) and miR-625-5p class (*p* = 0.005) (Appendix A). Finally, multivariate analyses with all the variables significant in the respective univariate models were conducted. For OR, the variables that retained their significance were PD-L1 staining (*p* = 0.025) and miR-625-5p class (*p* < 0.001) (Figure 4A). For OS, the variables that retained their significance were tumor burden (*p* = 0.002), PD-L1 staining (*p* = 0.028) and miR-625-5p class (*p* = 0.048) (Figure 4B). Variables that retained their significance in the multivariate PFS model were tumor burden (*p* = 0.001), PD-L1 staining (*p* = 0.004) and miR-625-5p class (*p* = 0.039) (Figure 4C).

### 3.4. Prognostic Classes Based on miR-625-5p and PD-L1

In the multivariate analyses, PD-L1 staining and miR-625-5p class were independently associated to patient outcomes. In order to further refine the prognostic classification, four groups based on the expression of miR-625-5p and PD-L1 (i.e., miR-625-5p High/PD-L1 < 50%, *n* = 32; miR-625-5p Low/PD-L1 ≥ 50%, *n* = 20; miR-625-5p High/PD-L1 ≥ 50%, *n* = 24; miR-625-5p Low/PD-L1 < 50%, *n* = 13) were compared.

Considering tumor size change at best response, the miR-625-5p Low/PD-L1 ≥ 50% group showed a mean reduction in tumor size of 34.45% (median 33.5%), whereas the other three groups showed a mean increase in tumor size ranging from 8.62% (median 13.5%) to 32.13% (median 30%) (*p* < 0.001) (Figure 5). Pairwise comparisons reveal that miR-625-5p expression (high vs low) was significantly associated with tumor shrinkage in patients with PD-L1 ≥ 50%, but not in PD-L1 < 50% groups.

In the OS analysis, the miR-625-5p Low/PD-L1 ≥ 50% group showed the longest median survival (27.0 months), the miR-625-5p High/PD-L1 ≥ 50% had an intermediate survival (10.75 months), whereas the groups with PD-L1 < 50% had the shortest one (6.3 months) independently from miR-625-5p status. These differences were statistically significant (Figure 6A). Similarly, in the PFS analysis, the miR-625-5p Low/PD-L1 ≥ 50% group showed the longest median survival (24.0 months), the miR-625-5p High/PD-L1 ≥ 50% had an intermediate survival (7.45 months), whereas the groups with PD-L1 < 50% had the shortest one (<4 months) both in High and Low miR-625-5p. In addition, these differences were statistically significant (Figure 6B). These data demonstrated that miR-625-5p status significantly influence both OS and PFS in patients with PD-L1 ≥ 50%.

### 3.5. In Silico Evaluation of Potentially Relevant miR-625-5p Targets

In order to identify mRNA putative targets of miR-625-5p, three tools were used (TargetScan 7.2, miRDB, DIANA-microT-CDS) [18,19,20]. TargetScan7.21, a sequence-based tool, retrieved 5107 predicted mRNA targets ranked according to a specific score, namely cumulative weighted context++ score. DIANA-microT-CDS2, an energy-based tool, retrieved 830 predicted mRNA targets ranked according a specific score, namely miTG score. miRDB3, a machine-learning-based tool, retrieved 937 predicted mRNA targets ranked according to a specific score, namely target score. More than 400 mRNAs were identified as potential targets by all programs, as shown by the Venn diagram (Appendix A). Putative mRNA targets were ranked by each tool according to their probability to bind miR-625-5p, and a cumulative ranking score was then generated using geometric mean. The top 50 genes are shown in Appendix A. Moreover, gene set enrichment analysis [21] showed a significant overrepresentation of genes involved in immune-system adaptative response (Appendix A).

## 4. Discussion

Accumulating evidence showed the clinical relevance of circulating free and EV-miRNAs as prognostic/predictive biomarkers of response to anticancer treatments in different types of solid tumors [22,23,24,25,26,27,28]. However, EV represents a better source of miRNAs for biomarker studies in terms of quantity, quality and stability of EV-encapsulated miRNAs compared to circulating free miRNA [29,30,31,32].

Our work identified pretreatment levels of EVs miR-625-5p as a specific biomarker associated to OR, OS and PFS in ICI-treated NSCLC patients. Although miR-625-5p showed a significant correlation with PD-L1 staining, it was able to identify patients who did not benefit from ICI despite their high PDL-1 expression. The objective and design of this study were specifically chosen to identify potential EV-miRNA candidates of both biological and clinical interest. As one of the aims was to explore novel potential mechanisms of resistance to ICIs in NSCLC patients, this study is exploratory in nature, and therefore does not include a confirmatory validation cohort [33].

Similar to the current study, Peng et al. also analyzed EV-miRNAs in NSCLC cancer patients treated with ICIs [12]. This study included a lower number of patients (9 patients) compared to our analysis (88 patients). In addition, they used differential centrifugation method to isolate exosomes, while we preferred ExoRNeasy kit, recently reported as a method that preferentially isolate EV-associated miRNA [34]. Finally, Peng et al. analyzed only tumor response, without including survival outcome information (OS and PFS) that we reported in our study. 

Differently from other studies, miRNA detection was performed here with the nCounter platform. Compared to other platforms, nCounter is known to yield a smaller fraction of miRNAs detected above the background [35]. This reduced sensitivity was likely at least in part responsible for the identification of few EV-miRNA candidates in this study. On the other hand, nCounter is provided with the higher specificity in term of miRNA cross-detection bias compared to sequencing and PCR-based methods [35].

Moreover, this platform has the advantage to eliminate potential bias associated with amplification, is widely available in clinical laboratories and has a short turnaround time, making it a platform with high potential clinical applicability. 

Recently, signatures based on free circulating miRNAs have also been developed to stratify NSCLC patient treatment with ICIs. In particular, a 24-miR signature classifier originally developed to predict lung cancer development and prognosis was independently associated to OR, PFS and OS in NSCLC patients treated with ICIs [10]. Contrarily to this study, the one presented here used an unsupervised approach to identify the best candidate EV-miRNAs with a sample size specifically calculated to consider both the radiological and the survival endpoints. A different 7-miR signature generated following unsupervised microRNA profiling was recently associated to OS only in NSCLC patients treated exclusively with nivolumab [9]. The different patients’ populations, and in particular the different source of miRNA (serum vs exosomes), might explain why miR-625-5p was not identified in this study. In fact, due to their structure, EV-miRNAs are more likely to be taken up by neighboring or distant cells, and they therefore are more likely to mechanistically modulate biological processes in target cells [17]. Data from The Cancer Genome Atlas showed that miR-625-5p is overexpressed in NSCLC samples (both squamous cell carcinoma and adenocarcinoma) compared to normal lung tissues [36]. Moreover, miR-625-5p has been shown to suppress inflammatory responses in human bronchial epithelial cells [37]. Intriguingly, gene set analysis revealed an enrichment of genes involved in adaptative immune response and the top predicted target in silico of miR-625-5p is GIMAP1 (GTPase of the immunity-associated protein 1), a known regulator of cellular and humoral immunity. GIMAP1 is intrinsically required to prevent mature T cells apoptosis in the periphery and it is critical for peripheral B cells’ survival [38,39]. 

Besides its independent prognostic role in ICI-treated NSCLC patients, the association of miR-625-5p levels also to OR suggests a potential role also as a predictive biomarker. However, this cannot be confirmed in absence of a group of patients not treated with ICIs. As the large majority of NSCLC patients were receiving ICIs either in first or subsequent lines at the time of accrual, renouncing to this control group allowed a larger sample size necessary for the current explorative analyses. 

The population enrolled in this study included patients with PD-L1 expression < 50%, who would not necessarily receive ICI monotherapy at the present time, but rather a combination of chemotherapy and pembrolizumab [40]. Although our findings do not directly translate to this subgroup of patients, ongoing studies are being conducted to also assess the predictive and prognostic role of miR-625-5p in this subgroup of patients. Notably, miR-625-5p identified patients with different outcomes in terms of OR and survival, particularly in patients with PD-L1 expression ≥50% who are routinely offered ICI monotherapy.

Of the 218 patients originally assessed for eligibility, 35 were excluded because of inadequate plasma samples and 8 died before the first CT scan restaging. The exclusion of patients who died before radiological evaluation represents a limitation, as the study did not report information regarding “fast progressors” (immortal time biases). Moreover, EV-miRNAs were only evaluated at the baseline and there is no longitudinal assessment of EV-miRNAs’ dynamics during treatment and upon progression. Finally, the limited EV characterization does not allow us to identify the major source of miR-625-5p (exosome vs. microvesicles). This goes beyond our scope that was to capture all EV-associated miRNas excluding circulating free miRNA. For this purpose, we used a spin column-based method for the isolation of total RNA from EVs that showed a high specificity for vesicular over nonvesicular RNA [41].

## 5. Conclusions

Overall, this study suggests that EVs-miR-625-5p might represent a novel biomarker to stratify patients affected by NSCLC treated with ICIs (nivolumab and pembrolizumab as monotherapy), in particular those with PD-L1 expression ≥50%. The association with OR and the biological association of miR-625-5p to immune-related processes also suggest a potential predictive role related to modulation of tumor-associated immunity. Confirmatory results in prospective validation studies involving PDL1 ≥ 50% NSCLC patients treated with ICIs as a first line of treatment could help translate this biomarker in clinical practice.

## Figures and Tables

**Figure 1 cancers-14-02435-f001:**
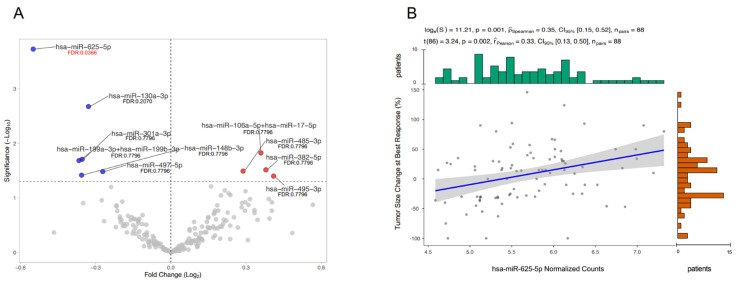
Differential EV-miRNAs’ expression based on the Objective Response. Volcano Plot shows miRNAs significantly overexpressed on the right (red dots) and miRNAs significantly downregulated on the left (blue dots) (**A**). Scatter plot representing the correlation between miR-625-5p levels and tumor size change at best response (**B**).

**Figure 2 cancers-14-02435-f002:**
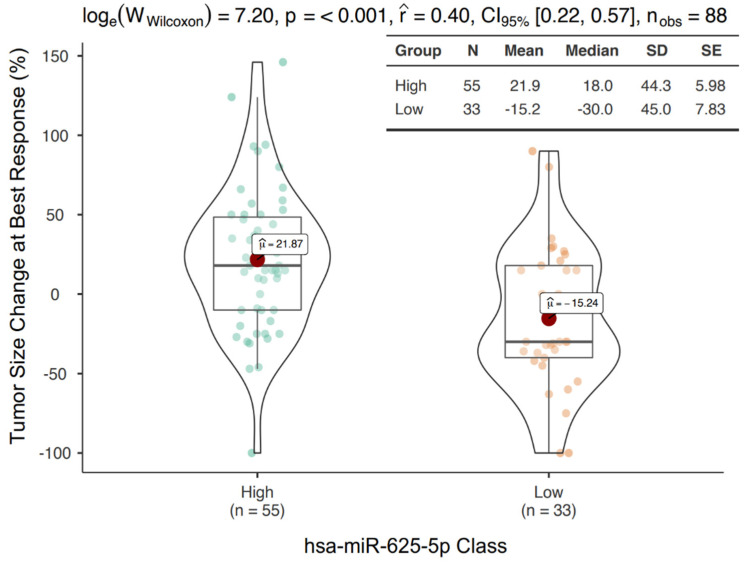
Tumor size changes according to miR-625-5p expression. Box/Violin plot representing the tumor size change at best response in miR-625-5p High and Low class.

**Figure 3 cancers-14-02435-f003:**
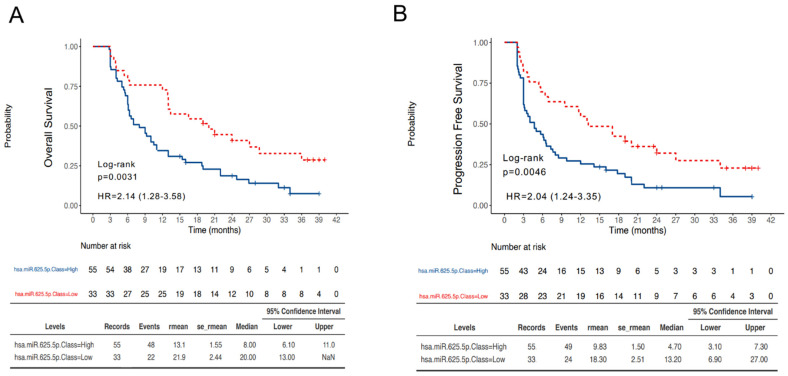
Survival outcomes according to miR-625-5p expression. Kaplan–Meier curves reporting the OS (**A**) and PFS (**B**) of patients dichotomized in miR-625-5p High and Low class.

**Figure 4 cancers-14-02435-f004:**
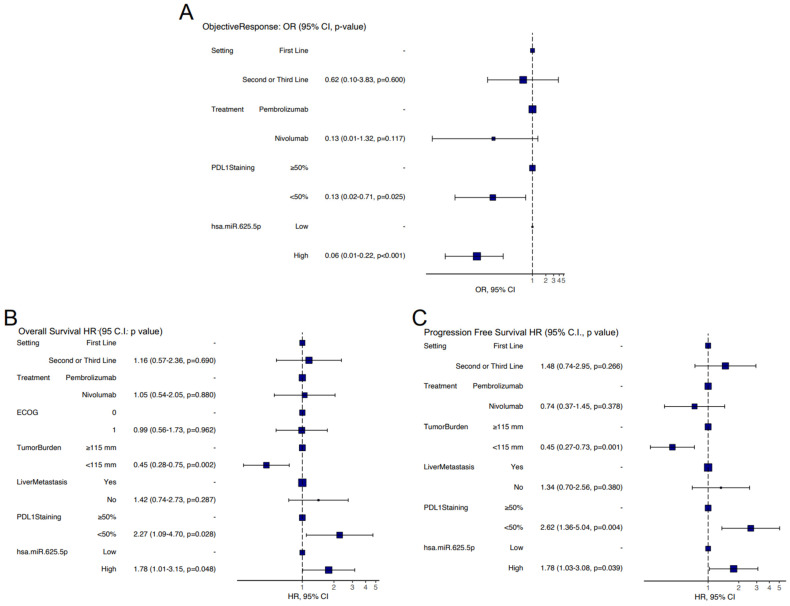
Multivariate analysis. Forest plot representing odd ratios and 95% C.I. for OR (**A**) hazard ratio and 95% C.I. for OS (**B**) and PFS (**C**) in multivariate analysis.

**Figure 5 cancers-14-02435-f005:**
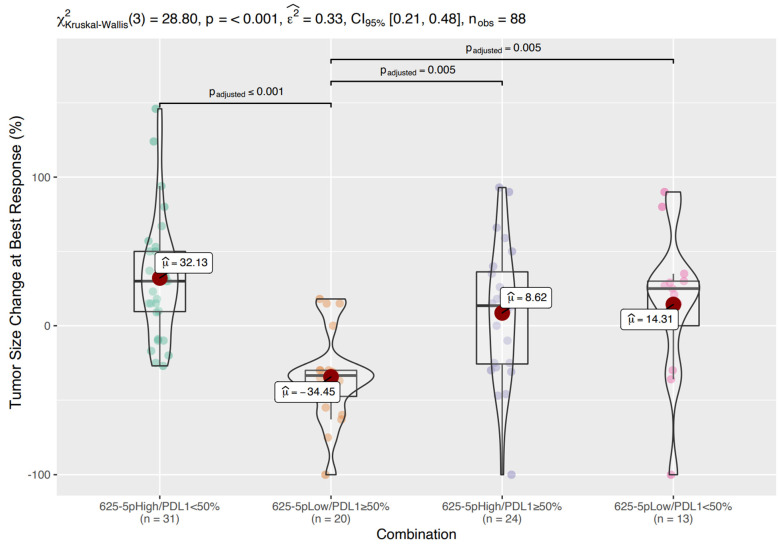
Tumor size changes according to miR-625-5p/PDL-1 expression Box/Violin plot representing the tumor size change at best response in the four classes of patients: miR-625-5p High/PDL1 < 50%, miR-625-5p Low/PDL1 ≥ 50%, miR-625-5p High/PDL1 ≥ 50% and miR-625-5p Low/PDL1 < 50%.

**Figure 6 cancers-14-02435-f006:**
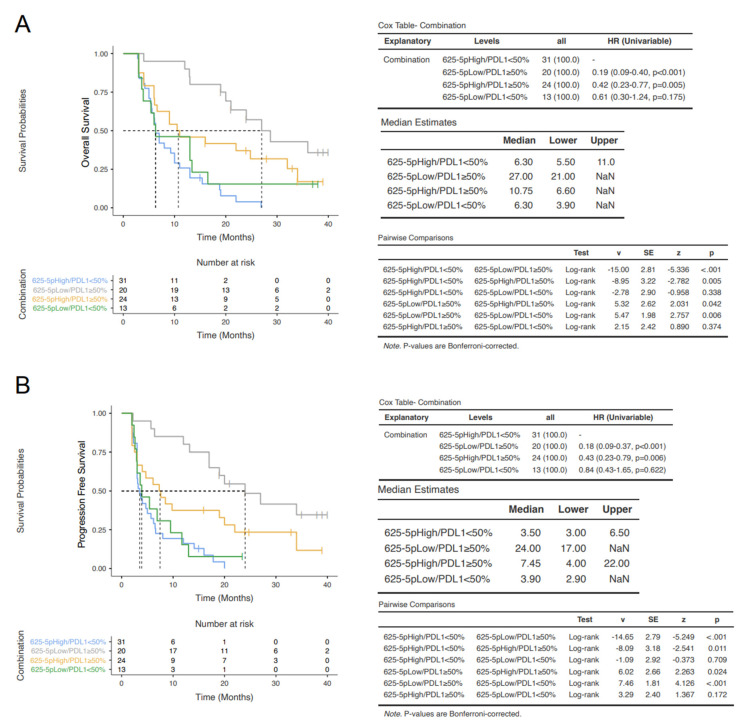
Survival outcomes according to miR-625-5p/PDL-1 expression. Kaplan–Meier curves reporting the OS (**A**) and PFS (**B**) in the four classes of patients: miR-625-5p High/PDL1 < 50%, miR-625-5p Low/PDL1 ≥ 50%, miR-625-5p High/PDL1 ≥ 50% and miR-625-5p Low/PDL1 < 50%.

**Table 1 cancers-14-02435-t001:** Descriptive statistics of clinic-pathological variables of study population.

	Overall (N = 88)
**Sex**	
Female	27 (30.7%)
Male	61 (69.3%)
**Age**	
≥70	42 (47.7%)
<70	46 (52.3%)
**Never Smokers**	
yes	6 (6.8%)
no	82 (93.2%
**ECOG**	
0	40 (45.5%)
1	48 (54.5%)
**Treatment**	
Nivolumab	35 (39.8%)
Pembrolizumab	53 (60.2%)
**Setting**	
Second or Third Line	51 (58.0%)
First Line	
**Histology**	
Adenocarcinoma	63 (71.6%)
Squamous Cell Carcinoma	25 (28.4%)
**EGFR**	
Mutated	4 (4.5%)
Wild Type	84 (95.5%)
**PDL1 Staining**	
<50%	44 (50.0%)
≥50%	44 (50.0%)
**Primary Tumor**	
Not Resected	63 (71.6%)
Resected	25 (28.4%)
**Tumor Burden**	
≥115 mm	45 (51.1%)
<115 mm	43 (48.9%)
**Brain Metastasis**	
Yes	19 (21.6%)
No	69 (78.4%)
**Liver Metastasis**	
Yes	17 (19.3%)
No	71 (80.7%)
**Lung Metastasis**	
Yes	54 (61.4%)
No	34 (38.6%)
**Bone Metastasis**	
Yes	52 (59.1%)
No	36 (40.9%)
**Pleural Effusion**	
Yes	17 (19.3%)
No	71 (80.7%)
**Adrenal Metastasis**	
Yes	72 (81.8%)
No	16 (18.2%)
**Soft Tissue Metastasis**	
Yes	6 (6.8%)
No	82 (93.2%)
**Nodel Metastasis**	
Yes	69 (78.4%)
No	19 (21.6%)
**Other Site Metastasis**	
Yes	11 (12.5%)
No	77 (87.5%)

## Data Availability

All data generated or analyzed during this study are included in the article and/or in the Appendix A.

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
