# Peer review of "Large-Scale Profiling of Extracellular Vesicles Identified miR-625-5p as a Novel Biomarker of Immunotherapy Response in Advanced Non-Small-Cell Lung Cancer Patients"

_cancers, 2022, doi:10.3390/cancers14102435_

Round 1

Reviewer 1 Report

The research paper written by Pantano et al. is very interesting. The statistical analysis is fine with a good cohort of patient.  The clinicians could identify, through your work, the patients who will have little or no benefit from ICI therapy despite the high PDL-1 expression.

Major comments.

None

Minor comments.

I suggest to the authors to have a better image quality in figure.

Please, correct the word microvesicles line 414; the word variable lines 458 and 459.

In the conclusion, please, precise the type of treatment to be combined (or not) with ICIs in case of PDL1>50% and miR-625-5p high. (An exemple of a concrete clinical application would be appreciated). 

Author Response

The research paper written by Pantano et al. is very interesting. The statistical analysis is fine with a good cohort of patient. The clinicians could identify, through your work, the patients who will have little or no benefit from ICI therapy despite the high PDL-1 expression.

Major comments.
None

Minor comments.
I suggest to the authors to have a better image quality in figure.
Reply: we provided to improve images quality

Please, correct the word microvesicles line 414; the word variable lines 458 and 459.
Reply: we provided to correct the mistakes

In the conclusion, please, precise the type of treatment to be combined (or not) with ICIs in case of PDL1>50% and miR-625-5p high. (An exemple of a concrete clinical application would be appreciated).
Reply: we specified in conclusion section that patients were treated with pembrolizumab or nivolumab as monotherapy

Reviewer 2 Report

In their manuscript „Large-scale profiling of extracellular vesicles identified miR-625-5p as a novel biomarker of immunotherapy response in advanced non-small cell lung cancer patients“ Pantano et al report analysis of micro RNA from extracellular vesicles (EV) obtained in a cohort of 88 out of 218 NSCLC patients who received anti-PD1 treatment. The authors applied strict inclusion criteria and analyzed the presence of a panel of 799 miRNAs. Tey subsequently performed extensive and rigorous statistical analyses revealing that exclusively miR625-5p suffices as independent prognostic biomarker.

Despite obvious deficiencies in the manuscript (the finding is limited, experimental or prospective validation is missing), this reviewer recommends that the work be accepted for publication provided that minor changes have been made.

Specific points:

Figures must be improved in quality and regarding labeling:

  • Individual charts in figures deserve a title
  • axis labels should designate the depicted variable
  • in figure legends skip “panel” and just indicate “a)” or “b)”

line 23:               grammar: repeatedly occurring throughout the text)

line 35-36:          what is “biological” or “clinical” significance? How can a biomarker capture interaction?

Line 45-46:         add e.g. “abundance of” miRNA625-5p was found to be correlated

Line 80: EV-miRNA profiles – delete “s” in “EV-miRNAs”

Line 108:            “(deaths)ranged” insert space

Line 131-134:    It seems a verb is missing and a full stop is missing before “Progression free survival”

Line 138:            “did not passed” – delet “ed”

Line 145-148:    redundant description – please simplify

Line 163-165:    rephrase, sentence is incomplete

Line 172:            Typo

Line 181:            delete “)”

Line 190-191:    rephrase, a variable is unlikely to be “significant” rather then e.g. “significantly different in group a as compared to group b).

Line 232:            “Panel A).Scatter” – space missing

Figure 2:             some obsolete pixels above “has-miR-625-5p Class”

Figure 3:             please add chart title, correct axis label

Line 262:            delete “described”

Figure 4:             please improve labelling

Line 290:            odd sentence

Line 302:            delete “o” from Violino

Line 315-316:    This appears to be the major finding and deserves more confidence

Figure 6:             Quality of text is low. Would it be possible to include text rather than converting text into pixels?

Line 324:            rephrase “software”, e.g. programmes, tools, packages, ….

Line 331:            shown rather than showed

Line 332:            Unclear “Targets mRNAs wereranked” – please rephrase

Line 343-344:    odd sentence “more investigated” - please rephrase

Line 344:            “to not exclude” equals “include”

Line 355-363:    The authors should directly compare specific features of studies.

Line 386:            odd: “more expressed”

Line 395             change “suggest” to “suggests”

Line 410:            unclear: “…limited by this detection and immortal time biases“ - please rephrase

Supplementary table 6: *) abbreviations not explained;  
**) Lines 5&6: double “Current Smoker”
***) “Node Met” – please use explicit phrases

Author Response

In their manuscript „Large-scale profiling of extracellular vesicles identified miR-625-5p as a novel biomarker of immunotherapy response in advanced non-small cell lung cancer patients“ Pantano et al report analysis of micro RNA from extracellular vesicles (EV) obtained in a cohort of 88 out of 218 NSCLC patients who received anti-PD1 treatment. The authors applied strict inclusion criteria
and analyzed the presence of a panel of 799 miRNAs. Tey subsequently performed extensive and rigorous statistical analyses revealing that exclusively miR625-5p suffices as independent prognostic biomarker.
Despite obvious deficiencies in the manuscript (the finding is limited, experimental or prospective validation is missing), this reviewer recommends that the work be accepted for publication provided that minor changes have been made.
Specific points:

Figures must be improved in quality and regarding labeling:
Individual charts in figures deserve a title
Reply: we provided to include a title for each figure.

axis labels should designate the depicted variable
Reply: we checked it in all figures

in figure legends skip “panel” and just indicate “a)” or “b)”
Reply: we deleted “panel” in the text

line 23: grammar: repeatedly occurring throughout the text)
Reply: we checked grammar across the text

line 35-36: what is “biological” or “clinical” significance? How can a biomarker capture
interaction?
Reply: we modified this sentence to clarify the meaning

Line 45-46: add e.g. “abundance of” miRNA625-5p was found to be correlated
Reply: we added “abundance of” as suggested

Line 80: EV-miRNA profiles – delete “s” in “EV-miRNAs”
Reply: we provided to delete it

Line 108: “(deaths)ranged” insert space
Reply: we provided to correct.

Line 131-134: It seems a verb is missing and a full stop is missing before “Progression free survival”
Reply: we provided to add a full stop.

Line 138: “did not passed” – delet “ed”
Reply: we provided to make it

Line 145-148: redundant description – please simplify
Reply: we simplified the sentence

Line 163-165: rephrase, sentence is incomplete
Reply: we provided to complete the sentence.

Line 172: Typo - Reply: we provided to correct it

Line 181: delete “)”
Reply: we provided to delete it

Line 190-191: rephrase, a variable is unlikely to be “significant” rather then e.g. “significantly different in group a as compared to group b).
Reply: we modified as suggested

Line 232: “Panel A).Scatter” – space missing
Reply: we added the space

Figure 2: some obsolete pixels above “has-miR-625-5p Class”
Reply: we modified the figure

Figure 3: please add chart title, correct axis label
Reply: we modified the figure

Line 262: delete “described”
Reply: we provided to delete it

Figure 4: please improve labelling
Reply: we improved it

Line 290: odd sentence
Reply: we modified the sentence

Line 302: delete “o” from Violino
Reply: we provided to delete it

Line 315-316: This appears to be the major finding and deserves more confidence
Reply: we modified the sentence

Figure 6: Quality of text is low. Would it be possible to include text rather than converting text into pixels?
Reply: we improved it

Line 324: rephrase “software”, e.g. programmes, tools, packages, …
Reply: we provided to rephrase “software”

Line 331: shown rather than showed
Reply: we provided to correct it

Line 332: Unclear “Targets mRNAs wereranked” – please rephrase
Reply: we modified the sentence

Line 343-344: odd sentence “more investigated” - please rephrase
Reply: we modified the sentence

Line 344: “to not exclude” equals “include”
Reply: we modified the sentence

Line 355-363: The authors should directly compare specific features of studies.
Reply: we modified this paragraph directly comparing specific features

Line 386: odd: “more expressed”
Reply: we modified the sentence

Line 395 change “suggest” to “suggests”
Reply: we provided to correct it

Line 410: unclear: “…limited by this detection and immortal time biases“ - please rephrase
Reply: we modified the sentence

Supplementary table 6: *) abbreviations not explained;
**) Lines 5&6: double “Current Smoker”
***) “Node Met” – please use explicit phrases
Reply: we modified the supplementary table

Reviewer 3 Report

The article presented for review is a good example of valuable research study, very modern and of importance for science and clinical practice. It is evidence that there is a need for search for  biomarkers and predictive factors for solid tumors  immunotherapy. Identification of EVs and miRNA seems to help in this however, with low chance to routine use.  Thus, for better present the results and to facilitate  its understanding I suggest some corrections:

  • Introduction- please explain in some sentences how : “ circulating EV-associated miRNAs (EV-miRNAs) could provide relevant information regarding not only cancer cell biology, but also tumor microenvironment including tumor-immune system interactions”, please cite some of the works of Whiteside T group.
  • Paragraph- Sample size- Here I need a help of statistician to state if such a plan brings a risk of not objective qualification and or if allow to perform an objective study?
  • 2- please correct this paragraph by division to patients and methods, separately. The inclusion criteria should be presented. Line 121- “inclusion criteria are patients…”- no, please correct. The table with patients characteristic should be presented in the main document, not supplement. Please show what was the stage of lung cancer.
  • My main doubt concerns the influence of previous treatment on the results. I suggest to clearly explain that you are sure that only ICIs are responsible for the results.

Author Response

The article presented for review is a good example of valuable research study, very modern and of importance for science and clinical practice. It is evidence that there is a need for search for biomarkers and predictive factors for solid tumors immunotherapy. Identification of EVs and miRNA seems to help in this however, with low chance to routine use. Thus, for better present the
results and to facilitate its understanding I suggest some corrections:

Introduction- please explain in some sentences how : “ circulating EV-associated miRNAs (EVmiRNAs) could provide relevant information regarding not only cancer cell biology, but also tumor microenvironment including tumor-immune system interactions”, please cite some of the works of Whiteside T group.
Reply: we provided to discuss briefly this point including some papers of Whiteside T research group.

Paragraph- Sample size- Here I need a help of statistician to state if such a plan brings a risk of not
objective qualification and or if allow to perform an objective study?
Reply: sample size was calculated to be adequately powered in order to perform an objective study (Response Rate and Overall Survival)

2- please correct this paragraph by division to patients and methods, separately. The inclusion criteria should be presented. Line 121- “inclusion criteria are patients…”- no, please correct. The table with patients characteristic should be presented in the main document, not supplement. Please show what
was the stage of lung cancer.
Reply: we provided to add a specific paragraph on inclusion and exclusion criteria. We moved the table with patients characteristics in main document

My main doubt concerns the influence of previous treatment on the results. I suggest to clearly explain that you are sure that only ICIs are responsible for the results.
Reply: according to multivariate analysis (figure 4), EV-miR-625-5p expression retains its performance in identifying patients who mostly benefit from ICI in term of objective response, overall survival and progression free survival independently from the treatment type and line.